# MLUTNet: A Neural Network for Memory Based Reconfigurable Logic Device Architecture

**Xuechen Zang and Shigetoshi Nakatake ***

The Graduate School of Environmental Engineering, Hibikino Campus, The University of Kitakyushu, Kitakyushu 808-0135, Fukuoka, Japan; z8dcb401@eng.kitakyu-u.ac.jp
* Correspondence: naka-lab@is.env.kitakyu-u.ac.jp

**Abstract:** Neural networks have been widely used and implemented on various hardware platforms, but high computational costs and low similarity of network structures relative to hardware structures are often obstacles to research. In this paper, we propose a novel neural network in combination with the structural features of a recently proposed memory-based programmable logic device, compare it with the standard structure, and test it on common datasets with full and binary precision, respectively. The experimental results reveal that the new structured network can provide almost consistent full-precision performance and binary-precision performance ranging from 61.0% to 78.8% after using sparser connections and about 50% reduction in the size of the weight matrix.

**Keywords:** approximate computing; memory reconfigurable logic device; binary neural network

## 1. Introduction

### 1.1. Neural Networks and Hardware Implementation

In many practical application scenarios, a large number of tasks do not have a complete logical description and do not require 100% absolutely correct implementation, such as tasks in the fields of visual recognition and natural language processing. High-speed and as accurate as possible results are clearly more attractive in these tasks than slow traditional manual processing that is close to full correctness.

A neural network is a machine learning (ML) technique that is inspired by and resembles the human nervous system and the structure of the brain. It consists of processing units organized in input, hidden, and output layers. The nodes of adjacent layers are connected to each other in a specific form and each has a weight value. The inputs are multiplied by their respective weights and added together at each node. The sum is transformed by an activation function and is then fed as input to the subsequent layers. The result of the final output layer is used as the solution to the problem. As one of the most widely used machine learning techniques in recent years, the neural network has achieved results on many complex cognitive tasks that match or far exceed human performance, making it a huge influence in a wide range of fields [1,2].

With the rapid growth of mobile and embedded devices in recent years, many implementations on hardware platforms have become feasible. Most implementations of neural networks in hardware involve two core components: storage and computing. The main part of the storage is the weight matrix of the neural network, the main body of which is a large number of floating point numbers, while the operations are dominated by matrix multiplication. Previously, a part of the work has been focused on limiting the accuracy of neural networks, such as ternary weight networks (TWN) [3], binary neural networks (BNN) [4], in favor of making them better deployment on hardware platforms with restricted computing power. The BNN series networks has gained more attention and extended research due to the great simplification of operations and its similarity to hardware circuit logic. Another part of the work demonstrates that accuracy-constrained neural networks can operate efficiently on hardware platforms, such as [5,6].

### 1.2. Memory Based Reconfigurable Logic Device

Recently, a kind of memory based reconfigurable logic device (MRLD) [7], taking multiple look-up tables (MLUT) as the core component structure, has demonstrated attractive benefits, such as low production cost, low power, and small delay, and programmable function to implement designed circuit logic. It is regarded as an promising alternation to FPGA in some respects. Its structure is shown in Figure 1.

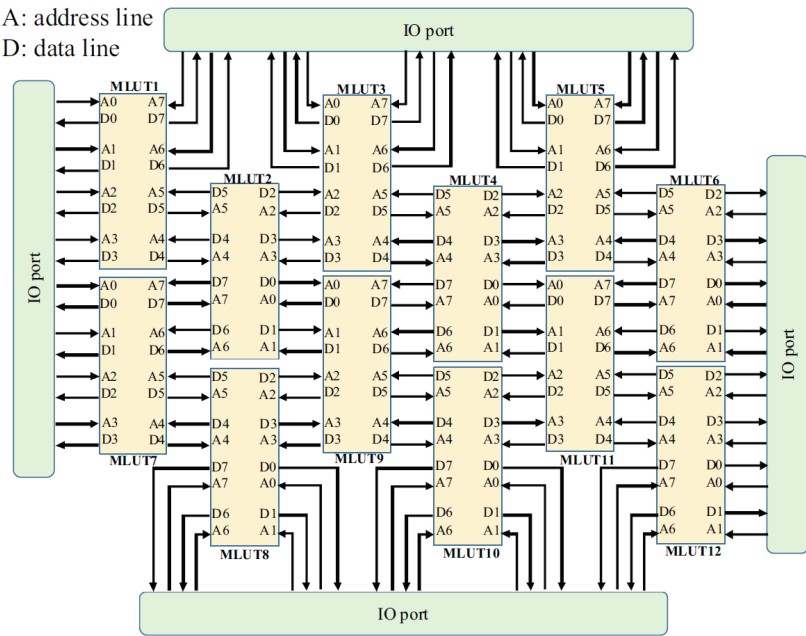

**Figure 1.** The structure of MRLD.

MRLD is an MLUT array constructed with a special internal connection structure. MLUTs are basic reconfigurable elements that consist of synchronous SRAM and asynchronous SRAM, with $m$-bit address inputs and $m$-bit data outputs forming a pair of interconnects. For an MLUT, the $m$-bit address input comes from the data outputs of its neighboring MLUTs, and the $m$-bit outputs are connected to the address inputs of these MLUTs. Each MLUT can operate as a memory block or a logical wiring block.

Compared to logic and switch block independent FPGAs, the MRLD has the following advantages: each MLUT can be used as a logic block or wiring element by configuring the corresponding truth table in the SRAM; the address input/data output of the MLUT will be the input and output of the configured logic circuit (or wiring element). Since the logic blocks and wiring elements are configured in the SRAM, there is no longer a need for as many interconnect resources as in an FPGA, making it possible to have high density reconfigurable devices with small latency and low power.

### 1.3. Motivations and Contributions

When deploying neural networks to MRLD-like programmable logic hardware, there are two most critical issues: storage cost and architecture conversion. Some works on limiting the numerical accuracy of neural networks have been presented, but how to structurally reorganize to cut the complexity and storage expense of the network is still a meaningful problem to be solved.

Based on the starting point of solving the above problem, we propose a new neural network based on the MRLD structure named MLUTNet. In MLUTNet, we adopt a network topology similar to the MRLD structure to partition the middle layer of the network in order to reduce the storage expense of neural network deployment on hardware platforms effectively. The contributions of this paper are as follows:

- We propose MLUTNet, a novel neural network with an atypical structure. MLUTNet combines the advantages of two aspects: the efficient learning performance of neural networks and the similar structure of MRLD, which makes MLUTNet easy to implement on MRLD or other similar logical storage devices without much extra effort;
- We conducted experiments and compared the results with their MLUTNet versions on three different popular datasets using standard and binary neural networks as the baselines, respectively. Compared to a fully connected neural network of the same size, MLUTNet saves over 50% of the weight matrix storage space and also reduces the size of individual weight matrices efficiently, with acceptable performance loss in accuracy on the dataset;
- The method is simple and easy to implement with good scalability; the MLUTNet version of a particular network can be substituted for the original network at no additional cost to meet the need to reduce the size of the network. This is very friendly for subsequent extension studies.

*1.4. Organization of the Paper*

The paper is organized as following:

- Section 2 presents the background about neural networks and the techniques used in this paper;
- Section 3 presents an explanation of the proposed MLUTNet and its operation principle;
- Section 4 shows the experimental results and analysis;
- Section 5 concludes and summarizes.

## 2. Backgrounds

*2.1. Low-Precision Neural Networks*

The neural network, is a popular structure used in approximate computing and machine learning. In a neural network, the basic operation that occurs layer by layer can be expressed as the following equation:

$$z = \sigma(w \times a + b) \tag{1}$$

where Equation (1) represents deep neural network mechanism. Variables z, w, a, and b represent the output tensor, the weight tensor, the activation tensor generated by the previous network layer and the bias tensor, respectively. $\sigma$ is the non-linear activation function. The main component of the forward propagation process is matrix multiplication. The backward propagation, on the other hand, is a chain law operation that calculates the gradient of each weight parameter relative to the final loss function. This leads to a problem of computational cost.

Assuming full precision for all parameters, the computational expense of a large number of matrix multiplications would be a non-negligible obstacle for implementation in hardware. Therefore, many works on low-precision neural networks have been proposed, the most representative of which is the binarized neural network (BNN) [8].

Compared with the full-precision general neural networks, the BNN makes two major changes: (1) 1-bit representation is utilized to replace the full-precision parameters. To guarantee correct gradient descent process, the gradient values and not-well-trained weight values are still calculated and stored in full-precision; but after the training process, the weight values are stored in 1-bit binary value; (2) Complex floating multiplication operations of weight values and activation values are replaced by XNOR-Bit-count operations, as the Equation (2). This could save significant amount of resource of computation in hardware implementation.

$$a = \mathrm{popcount}\left(\mathrm{xnor}\left(a^b, w^b\right)\right) \tag{2}$$

As a result, although the BNN inevitably suffer from the performance degradation, because of largely saving the parameter complexity and computing cost, it is still regarded

as a promising technique for deploying deep models on resource-limited devices. Many related works have flourished, for example, the XNORNet [9] succeeds to extend it to the convolutional neural network structure.

### 2.2. Optimizer and Learning Rate Scheduler

For the training of neural networks, the settings of optimizer and learning rate scheduler are crucial. Compared with the universal stochastic gradient descent (SGD) method, RMSProp [10] achieves a better balance between convergence speed and accuracy, and has a stronger ability to get out of the saddle points.

The learning scheduler is usually set to an exponential decay type. Several works have investigated the effect of learning rate variation on network performance. One of the representative works is the cosine annealing + Warm-Up [11]. The learning rate is increased or decreased periodically according to the cosine function and each period is longer than the previous one, depending on the proportion of the warm-up parameter. The study shows that the cosine annealing learning scheduler is more likely to achieve better results on many datasets.

### 3. MLUTNet

### 3.1. Network Definition

Assuming an end-to-end approach to train directly on full precision neural networks, i.e., using the input-target data pairs of data as training data and then mapping the trained neural networks to MRLD, we will face many potential problems. (1) The hidden layers of standard neural networks are usually fully connected, the size of individual weight matrix is large, and the number of connections grows exponentially with the size of the weight matrix. From the SRAM storage point of view, it is unrealistic to achieve a reasonable mapping on MRLD without limiting the size of the weight matrix and the number of connections. (2) The parameter storage and arithmetic processes inside the standard neural network are executed at full precision by default. However, most programmable logic devices support limited precision, and the operational expenses increase dramatically when the precision is higher. Therefore, it is important to use a neural network model with limited precision to find a balance between accuracy and cost. (3) The non-linear activation functions commonly used in neural networks, such as *Sigmoid* and *tanh*, usually require proprietary resources for implementation on programmable logic devices and can only be approximated with limited accuracy. Although the expense of a single operation is not significant, considering that each parameter of the hidden layer weights is involved in the activation operation, a more sensible measure is to switch to other activation functions that are more friendly to hardware mapping.

To address the above issues, we use the following measures to improve them. For (1), we import the methods of splitting the hidden layer and neighbor-only connections. Since MRLD devices are composed of multiple MLUTs as basic cells, each MLUT can be regarded as a small SRAM array with data storage or logic wiring or both. We split the originally larger hidden layers according to the MRLD topology, perform the computation and parameter update separately, and merge them into the final result after the computation is completed. For the connections between the split weight matrices, we utilize neighbor connections to reduce the number of connections and also play a role in preventing over-fitting. For (2), we import the idea of low precision neural network, BNN, as the comparative alternative to the full-precision standard neural network. BNN uses a 1-bit representation for the weights and XNOR-bit operation instead of the usual multiplication operation. Compared with full-precision multiplication, both the software training cost and the hardware computation cost can be effectively reduced. For (3), we use the *ReLU* function [12] and *Htanh* functions as activation functions instead of the traditional *Sigmoid* and *tanh*. Benefiting from their simple numerical logic, both can be implemented by linear combinations of logic gates.

$$\text{ReLU}(x) = \max(0, x) \tag{3}$$

$$\text{Htanh}(x) = \max(-1, \min(1, x)) \tag{4}$$

### 3.2. Training and Connection of MLUTNet

Based on the above analysis, we propose a new artificial neural network structure, MLUTNet. Figure 2 shows an illustration of the conversion and comparison of a single hidden layer neural network with MLUTNet.

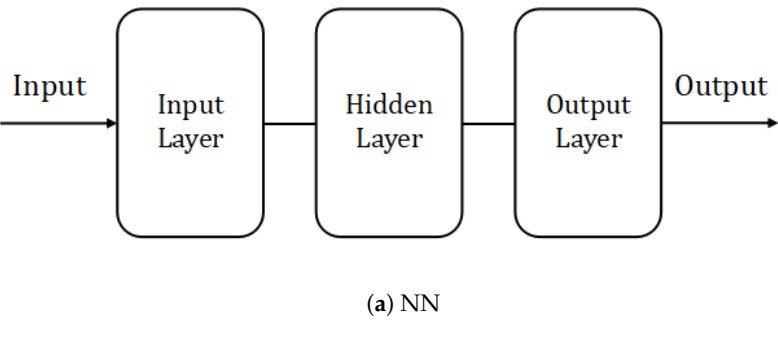

(**a**) NN

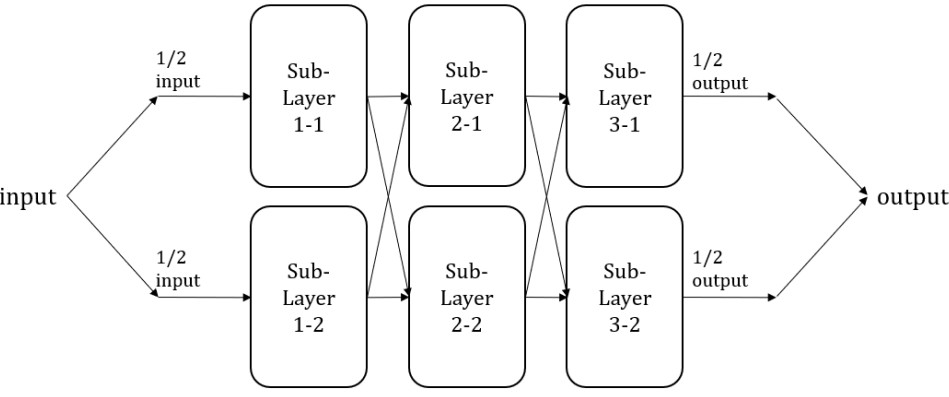

(**b**) MLUTNet

**Figure 2.** One NN and MLUTNet.

In addition to having the learning potential of a standard neural network, the weight matrix separated structure and sparse connection design employed by MLUTNet make it more friendly for subsequent mapping and implementation on MRLD and similar hardware. Based on the trained MLUTNet, the weight matrix can be mapped to MLUT units and wired on the MRLD depending on the network connectivity. If the split matrix still exceeds the unit storage limit of the device, then it needs to be stored separately in multiple memory units while increasing the cost of data exchange in the network implementation. It means that there will be multiple MLUTs that are combined as a larger generalized "Big-MLUT" within which the weight matrix is stored and data are exchanged.

A key point of interest is how each sub-layer should be connected to the next sub-layer between adjacent hidden layers. A plain and natural idea is to use full connectivity in the same way that neurons within a hidden layer are connected to each other. However, unfortunately, this approach is not feasible. On the one hand, it is difficult or impossible to fully interconnect the MLUT units storing the weight matrix due to the limitation of the number of connections within the hardware and cost considerations. On the other

hand, for MLUTNet, full interconnection between sub-layers does not necessarily enhance performance, but may cause degradation of network performance by reducing the sparsity of the network, as reflected in the experimental data in Section 4.

Considering the need to fit the structure of MRLD as closely as possible and to reduce the obstacles for subsequent implementations, we use neighborhood connection, as shown in Figure 3. The connections between sub-layers will be dropped by some determined logic (e.g., red connection are dropped in the odd number of layers and blue ones in even layers). If a sub-layer has input from more than one sub-layer, the input it receives is summed up as the new input. This connection method has several advantages: firstly, sparse neighbour-only connections make the network structure as similar as possible to the MRLD topology, reducing barriers to subsequent implementation; furthermore, sparse inter-layer connections reduce the possibility of over-fitting; and finally, a smaller number of connections reduces the computational and memory overhead of the network. The disadvantage is that each sub-layer can only affect its neighboring sub-layers, so the network needs to be deep enough to ensure learning capability.

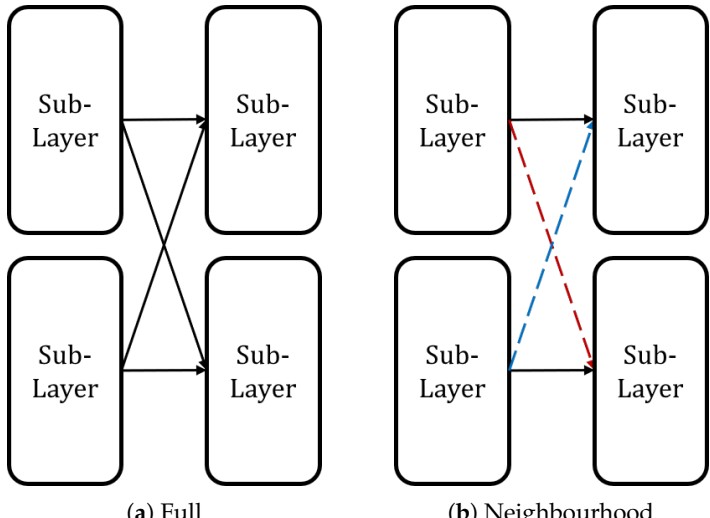

(**a**) Full        (**b**) Neighbourhood

**Figure 3.** Full connection and neighbourhood connection.

As a result of the above discussion, the network logic of MLUTNet was finalized as follows. Layers of the network are split into multiple sub-layers, and no connection is created between sub-layers of the same layer. The activation function is attached after each sub-layer. During the training process of the network, the training of sub-layers in the same layer is run in parallel. In the input layer, the input data matrix is cut equally along the Y-axis and used as the output of the sub-layers of the input layer, respectively. At the output layer, the outputs of the two sub-layers are combined along the Y-axis and used as the final output.

To make the process more understandable, we illustrate the process with a segment of actual MNIST data going through MLUTNet. As shown in Figure 4, the test data are a gray-scale image with an initial dimension of $28 \times 28$ and the content is a handwritten number whose label is a one-dimensional vector of size 10. This vector is one-hot, i.e., only one element is 1 and the rest is 0. The x-axis coordinate of element 1 represents the content of the image, i.e., which of the numbers 0 to 9 is the image. Suppose we use a batch size of 100 for each training, i.e., the initial dimension of the data is $100 \times 28 \times 28$. After binarizing the data, the size of the input data is $100 \times 784$. When entering the input layer, the input data are divided into two matrices of size $100 \times 392$. It is then multiplied with the weight matrix of size $392 \times 392$ in the hidden layer. In the output layer, we will finally get two matrices of size $100 \times 5$ and merge them into the final result of $100 \times 10$. For more specific process details, the MLUTNet generic training flow written according to the algorithm format is shown in Algorithm 1.

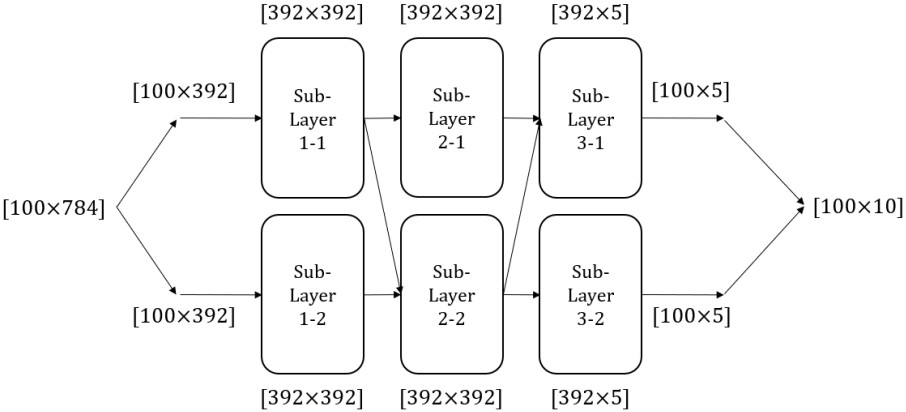

**Figure 4.** Dataflow in MLUTNet on MNIST.

---

**Algorithm 1** Training of MLUTNet.

---

**Require:** the number of network layers $L$, the number of sub-layers each layer $S$, weights of network $W$, activation values $a$, gradient values $g$, learning rate $\eta$
**Ensure:** updated weights $W^{t+1}$, updated learning rate $\eta^{t+1}$
1:  **{Forward propagation}**
2:  **for** $k$ in range$(1, L)$ **do**
3:    **for** $m$ in range$(1, S)$ **do**
4:     (if Binary) $W_{k,m} \leftarrow Binarize(W_{k,m})$
5:     $a_{k,m} \leftarrow a^b_{k-1,m} W_{k,m}$
6:     (if Binary) $a_{k,m} \leftarrow Binarize(a_{k,m})$
7:    **end for**
8:  **end for**
9:  **{Backward propagation}**
10:  **for** $k$ in range$(L, 1)$ **do**
11:    **for** $m$ in range$(1, S)$ **do**
12:     $g_{a_{k-1,m}} \leftarrow g_{a_{k,m}} W_{k,m}$
13:     $g_{W_{k,m}} \leftarrow g^{\top}_{a_{k,m}} a_{k-1,m}$
14:    **end for**
15:  **end for**
16:  **{Updating parameters}**
17:  **for** $k$ in range$(1, L)$ **do**
18:    $W^{t+1}_k \leftarrow Update(W_{k,m}, \eta, g_{W_{k,m}})$
19:    $\eta^{t+1} \leftarrow Scheduler(\eta)$
20:  **end for**

---

## 4. Experimental Results

We configure four types of network models, standard neural network(NN), binary neural network (BNN), MLUTNet neural network (MLUTNet), and binary MLUTNet (B-MLUTNet) neural network. The experimental results are operated under the following hardware environment: Intel i7-6700HQ, NVIDIA GTX 1060 and 16 GB RAM. The related codes are implemented by PyTorch framework.

Details configurations about the models are briefly described as follows. The ratio of training dataset to validation dataset is set as 80%:20%. RMSProp optimizer is used as the optimizer for the experiments. The epochs of the experiments are set to 40. The initial learning rate is 0.001. The learning rate scheduler is set to exponential decay mode and cosine annealing mode, respectively, the exponential decay mode halves the learning rate every 10 epochs, the cosine annealing mode period is set to 5, and the "Warm-up" multiplier parameter is set to 2.

### 4.1. Performance on MNIST Series Datasets

The experiments conducted on three MNIST series datasets: MNIST [13], K-MNIST [14], and fashion MNIST [15]. In these datasets, the contents of images in datasets are graphics of handwritten numbers, Japanese hiragana characters from ancient books, and fashion items, respectively. The images are gray-scale images with the resolution of $28 \times 28$. The accuracy convergence process and the corresponding confusion matrices for each group of experiments are exhibited in Figures 5 and 6.

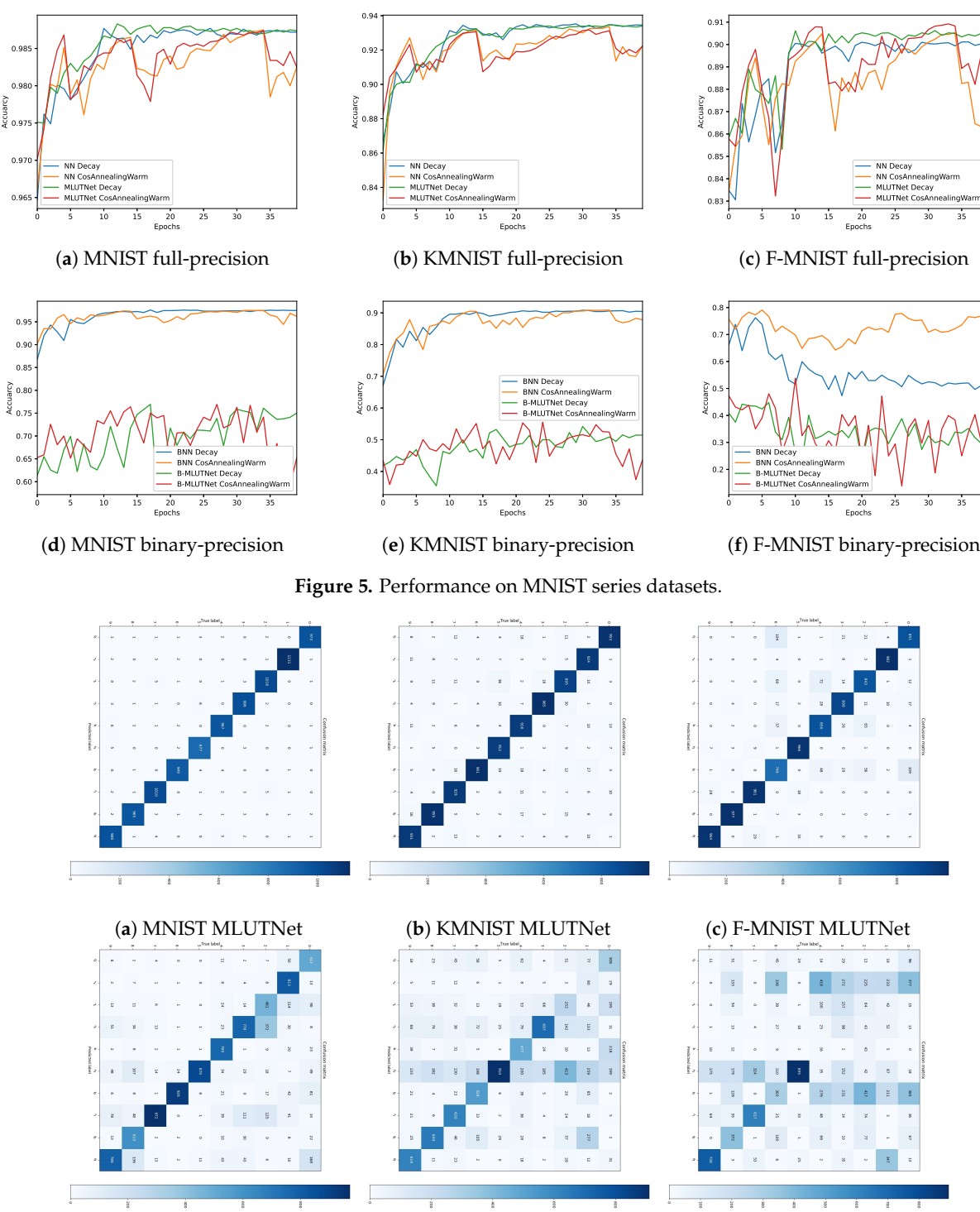

**Figure 5.** Performance on MNIST series datasets.

**Figure 6.** Confusion matrices of MLUTNet.

By reviewing the experimental results, some conclusions can be revealed. (1) In the full precision case, MLUTNet exhibits comparable performance to the standard NN on all three datasets. With enough epochs, the accuracy of the test dataset for each case converges steadily to approximately the same level. (2) In the binary accuracy case, MLUTNet has different degrees of performance loss compared to the standard BNN. It is the largest in KMNIST with 35.3% and the smallest in MNIST with 20.6%. (3) The performance of the decay scheduler is more stable, but cosine annealing shows better performance in F-MNIST. The full-accuracy MLUTNet with cosine annealing scheduler achieves a lead of about 0.71% over the NN with a decay scheduler; the binary-precision MLUTNet with Cosine annealing scheduler improves the accuracy by 9.03% over using the decay scheduler.

### 4.2. Performance on CIFAR-10 and STL-10 Datasets

In the CIFAR-10 dataset, images are divided into 10 categories. Each image is an RGB image with three color channels and a resolution of $32 \times 32$. In the STL-10 dataset, the resolution of the images is further improved to $96 \times 96$.

The experimental results on the CIFAR-10 dataset and the STL-10 dataset are shown in Figure 7. At full precision, the NN and MLUTNet achieve about 57% accuracy on the CIFAR-10 dataset and about 45% accuracy on the STL-10 dataset, which is a normal performance for standard structured neural networks. The confusion matrices are shown in Figure 8. However, at binary precision, BNN and binary MLUTNet cannot learn and converge smoothly due to the limited structural complexity.

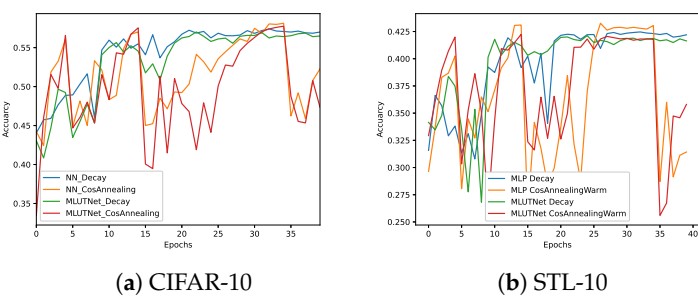

| (a) CIFAR-10 | (b) STL-10 |
|:---:|:---:|

**Figure 7.** Performance on CIFAR-10 and STL-10 dataset.

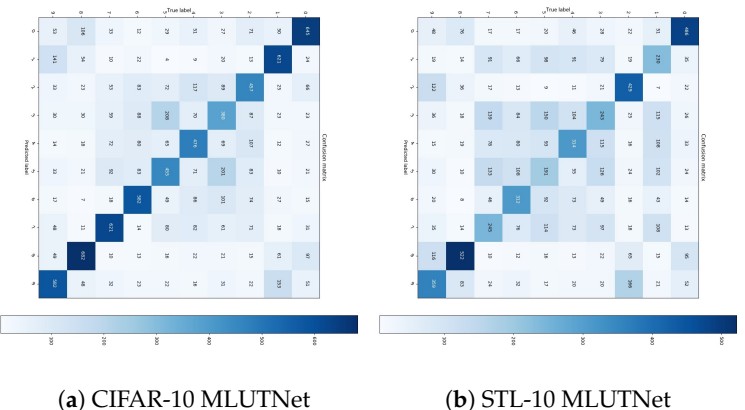

| (a) CIFAR-10 MLUTNet | (b) STL-10 MLUTNet |
|:---:|:---:|

**Figure 8.** Performance on CIFAR-10 and STL-10 dataset.

### 4.3. Sparsely Connection Verification

We conducted a set of comparative experiments regarding the way neighboring sub-layers are connected in MLUTNet. In the experiments, MLUTNet and Binary-MLUTNet are connected according to fully connected and sparsely connected, respectively, and tested under the same dataset, and the results are shown in Figure 9.

The fully connected MLUTNet does not show any significant advantage while increasing the computational effort, while the fully connected B-MLUTNet causes a significant

decrease in accuracy instead. We speculate that this is because the sparsely connected sub-layers somehow avoids premature over-fitting.

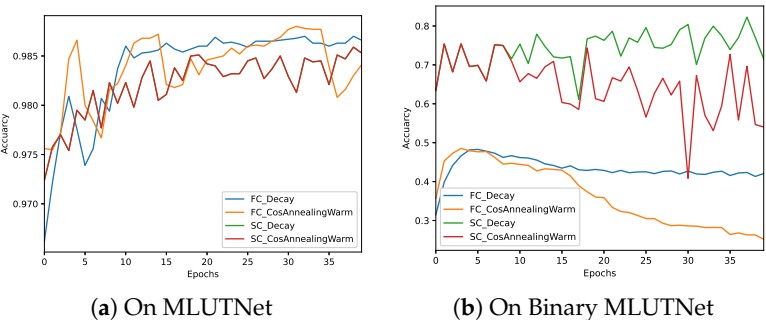

(**a**) On MLUTNet                  (**b**) On Binary MLUTNet

**Figure 9.** Comparison of fully connection and sparsely connection.

*4.4. Results Summary*

A summary of the optimal results of different accuracy models under each model is shown in Table 1. The run-time of models are shown in Figure 10.

**Table 1.** Optimal accuracy performance.

|            | NN     | MLUTNet | BNN    | B-MLUTNet |
|------------|--------|---------|--------|-----------|
| MNIST      | 0.9877 | 0.9883  | 0.9759 | 0.7693    |
| K-MNIST    | 0.9347 | 0.9348  | 0.909  | 0.5552    |
| F-MNIST    | 0.9053 | 0.9092  | 0.7909 | 0.5382    |
| CIFAR-10   | 0.5813 | 0.5775  | N/A    | N/A       |
| STL-10     | 0.4305 | 0.4186  | N/A    | N/A       |

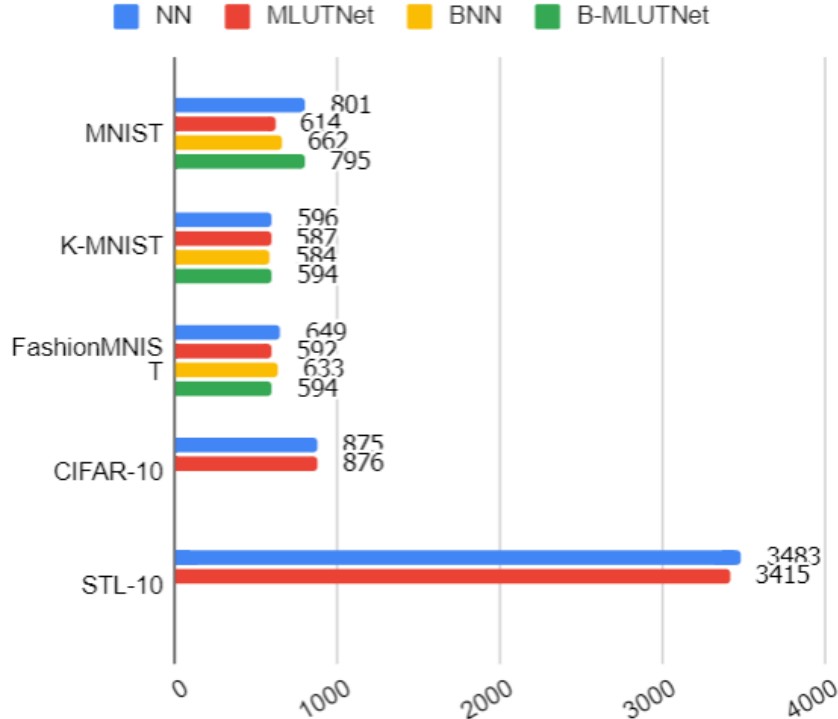

**Figure 10.** Models operation time.

On all four datasets, using the results of the standard NN as the baseline, Figure 11 shows the correct rate performance ratio of MLUTNet. The results show that the final accuracy performance of MLUTNet in full-precision is comparable to that of the standard NN model, and the size of the former's weight matrix is about half that of the latter; in binary-precision, depending on the dataset, respectively, the accuracy performance of MLUTNet ranges from 61% to 78.8% that of the standard BNN model.

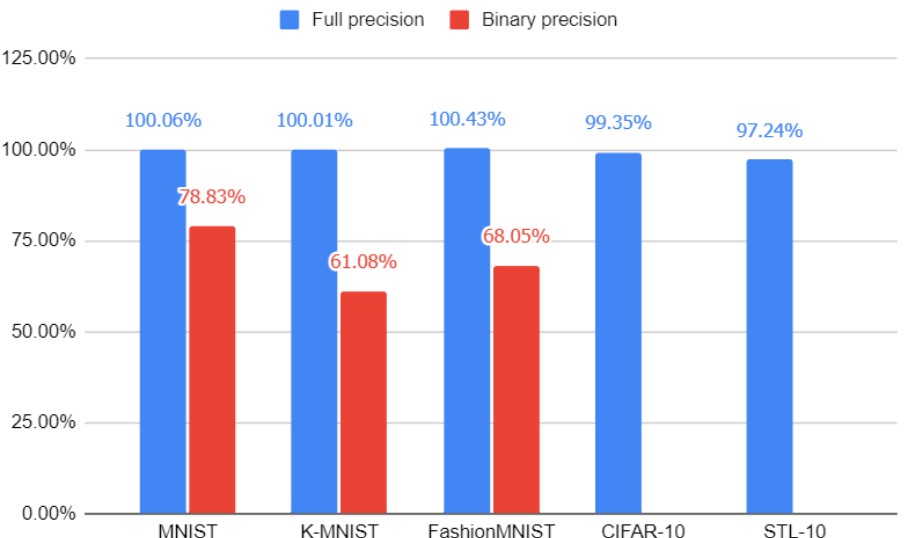

**Figure 11.** Correct Rate Performance Ratio of MLUTNet and NNs.

The experimental results demonstrate the effectiveness of MLUTNet. Under full-precision, MLUTNet achieves comparable performance to standard NN using a smaller scale weight matrix and fewer inter-layer connections; under binary-precision, MLUTNet suffers from the accuracy impact caused by the reduced weight matrix, and the final accuracy is reduced compared to standard BNN.

## 5. Conclusions

In this paper, we propose MLUTNet, an innovative neural network structure, by combining the structural features of MRLD devices. In MLUTNet, we import and effectively utilize measures, such as binarized weights, scale reduction in the weight matrix, and sparse connection, to significantly reduce the computing expense and porting cost of the network and maintain a relatively acceptable performance.

The experimental results reveal that MLUTNet achieves essentially equivalent or slightly better accuracy on test datasets. Moreover, on binary-precision networks, although cutting the weight matrix brings a greater additional precision impact, MLUTNet also achieves 78.8%, 61.0%, and 68.0% of the performance of the standard NN model on the three datasets of the MNIST series, respectively.

In summary, we have verified the effectiveness of MLUTNet in full-precision on general datasets, and further improving the performance and stability through optimization measures will be the focus of our next step. In addition, in this paper, due to the limitation of the standard NN's own structure, it cannot learn and converge smoothly on the CIFAR-10 dataset with binarization accuracy. Therefore, further extension of the MLUTNet structure to convolutional neural networks will also be a target of our future research.

**Author Contributions:** Conceptualization, X.Z. and S.N.; methodology, software, validation, formal analysis, investigation, resources, data curation, writing—original draft preparation and visualization, X.Z.; writing—review and editing, supervision, project administration and funding acquisition, S.N. All authors have read and agreed to the published version of the manuscript.

**Funding:** This research received no external funding.

**Data Availability Statement:** Not applicable.

**Conflicts of Interest:** The authors declare no conflict of interest.

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
