# Peer review of "MLUTNet: A Neural Network for Memory Based Reconfigurable Logic Device Architecture"

_applsci, doi:10.3390/app11136213_

Round 1
Reviewer 1 Report
In the paper author(s) presents a novel neural network and their synthesis scheme in memory-based programmable logic device. This new approach seems to be promising what is proved by the experimental results.
The paper is well written, ie. well structured and with correct English.
Line no. 88 on the page 3 seems to be incorect: indentation at the beginning is useless and the sense of the sentence is grammatically unclear.
Author Response
Thank you for your high praise for the content and writing of the article.
There was a text editing error in line 88 that you pointed out and we have fixed it in our latest submission.
Reviewer 2 Report
This paper proposes MLUTNet, a hardware-oriented neural network (NN) architecture with the enablement of tradeoffs between highly accurate inference and memory efficiency. The idea of the paper, i.e. co-exploration of hardware platform and NN architecture, is in line with the current trend in edge computing. The biggest challenge to mapping NN architecture to the memory-based reconfigurable logic devices (MRLD) is the SRAM storage overhead. By jointly splitting, quantizing, and sparsifying the weight matrices in hidden layers, experimental results show that MLUTNet has good performance in dealing with visual recognition tasks with a relatively low memory/computational overhead, compared to the standard NN architectures.
The following aspects of the paper are commendable:
- An easy-to-read language style is used.
- The manuscript is well-organized with proper abstract, introduction and conclusion, and has a clear algorithm explanation as well.
- The problem formulations and the corresponding contributions are clearly mentioned while highlighting the advancements they provide.
The topic is novel and impressive. However, some areas of the paper could be improved or need further explanation:
- Indeed, multiple look-up tables (MLUT) may be an alternative to FPGA; however, it is not entirely clear to me how the two properties mentioned in lines 39~42 of page 2 make MLUT more preferable. Under FPGA design, LUTs serve as both logic/memory components by mimicking logic gate combinations and constructing distributed RAM, respectively. Can MLUT achieve so more efficiently? It will be clearer if a comprehensive comparison between MLUT and FPGA is provided.
- Fig.1 is not entirely reflective of the function of MRLD. It gives the impression that MRLD only connects the data outputs of one MLUT to the address of another MLUT. Perhaps a clearer explanation of the function of MRLD, e.g. dataflow and common operations, can help the readers appreciate the structural features of this novel memory-based programmable logic device.
- It is not explained why the individual weight matrices are split into two equal-sized sub-matrices. I assumed that it is because each derived sub-matrix can be directly mapped on MRLD. Then, how does MLUTNet tackle the situation where splitting the weight matrices in half still leads to sub-matrices larger than the maximum size limit of MRLD? It would be good to clarify it in the text.
- There are few explanations of the rationale for the usage of sparse connection. For MLUTNet, the neighbor-only connections reduce the memory overhead, and the authors describe this technique as Dropout method. However, Dropout refers to stochastic computation during the training phase of NN. It seems that the neighbor-only connections are predefined; therefore, it would be better if the relationship between Dropout and neighbor-only connections could be explained a bit clearer.
- MLUTNet may have less time/storage complexity than standard NN architecture, both in the training and testing phase (This result is also mentioned in this paper). However, there are no experimental results about processing time and consumption of computing resources in this paper. Adding these contents can make this paper more complete.
- The experimental results are overstated. For example, the authors conducted experiments on MNIST-like and CIFAR-10 datasets to validate the effectiveness of MLUTNet. Since they are both relatively simple datasets, it is not surprising that simplifying the NN architectures leads to a negligible classification accuracy drop. Furthermore, fully-connected NN is used, instead of common Convolutional Neural Networks (CNN). Therefore, I think the results are not that convincing, and it is better to make some experiments that are based on other datasets and NN topologies.
- Another key criticism related to point 3 above is that B-MLUTNet in Table 1 can only reach 77% accuracy on the MNIST dataset. So, I think it is important to show the comparison of B-MLUTNet and other lightweight machine learning algorithms like LDA and SVM. A standard NN of INT8 precision is also an appropriate benchmark to be included in the experimental part. However, this paper only compares the proposed algorithm with standard BNN, so the advantages of B-MLUTNet are not clear. Maybe a more detailed explanation is needed.

Reviewer 3 Report
This is the review of the paper titled "MLUTNet: A Neural Network for Memory-based Reconfigurable Logic Device Architecture"
The paper is very well-written and presented. I would suggest some comments to improve it
1- In terms of the effectiveness of the proposed network, I would suggest testing it with more standard datasets.
2- Comparison with state-of-the-art methods is necessary to show the difference in the new network.
3- It is optional to give a brief intro about NN and Deep learning and I would suggest the following references
https://ieeexplore.ieee.org/abstract/document/8694781
https://link.springer.com/article/10.1186/s40537-021-00444-8
4- Confusion matrix values (TP,TN,...) have to be presented.
5- Vistulazation of filters of convolutional layers is necessary.
6- Training parameters including learning rate, optimizer, etc are missing.
7- In regard to open science, authors could make their implementations available through git or other repositories.
8-References are old.
Round 2
Reviewer 2 Report
The paper quality has substantially improved after the revision. There are just two points that are not clear to me, as illustrated below.
- Does Fig. 10 report the model operation time of a single inference or the entire (testing) dataset?
- 11 caption is misleading. It gives an impression that numbers are the reported accuracies of full-precision MLUTNet and binary MLUTNet (but they are the ratios to the accuracy of the standard NN?).
Author Response
Does Fig. 10 report the model operation time of a single inference or the entire (testing) dataset?
A: The reported operation time is the "training time used by the model to reach the optimal accuracy in 40 epochs on this dataset", and the time spent on the validation and test sets was not added.
Fig. 11 caption is misleading. It gives an impression that numbers are the reported accuracies of full-precision MLUTNet and binary MLUTNet (but they are the ratios to the accuracy of the standard NN?).
A: Yes, they are ratios compared to the accuracy of standard NNs. The caption is "Correct Rate Performance Ratio" and what we actually want to say is "Correct Rate Performance Ratio of MLUTNet and NN". We will modify it in the final submission.
Reviewer 3 Report
Comments very well addressed
Author Response
Thank you for your recognition of the revision.